# Nanomaterials and the Serosal Immune System in the Thoracic and Peritoneal Cavities

**DOI:** 10.3390/ijms22052610

**Published:** 2021-03-05

**Authors:** C. Frieke Kuper, Raymond H. H. Pieters, Jolanda H. M. van Bilsen

**Affiliations:** 1Consultant, Haagstraat 13, 3581 SW Utrecht, The Netherlands; 2Immunotoxicology, Institute for Risk Assessment Sciences, Utrecht University, Yalelaan 1, 3584 CL Utrecht, The Netherlands; raymond.pieters@hu.nl; 3Innovative Testing in Life Sciences & Chemistry, Research Centre for Healthy and Sustainable Living, University of Applied Sciences Utrecht, Padualaan 97, 3584 CH Utrecht, The Netherlands; 4Department for Risk Analysis for Products in Development, Netherlands Organization for Applied Scientific Research (TNO), Princetonlaan 6, 3584 CB Utrecht, The Netherlands

**Keywords:** innate lymphocytes, FALC, MS, pericardium, peritoneum, pleura, SALC, serosa

## Abstract

The thoracic and peritoneal cavities are lined by serous membranes and are home of the serosal immune system. This immune system fuses innate and adaptive immunity, to maintain local homeostasis and repair local tissue damage, and to cooperate closely with the mucosal immune system. Innate lymphoid cells (ILCs) are found abundantly in the thoracic and peritoneal cavities, and they are crucial in first defense against pathogenic viruses and bacteria. Nanomaterials (NMs) can enter the cavities intentionally for medical purposes, or unintentionally following environmental exposure; subsequent serosal inflammation and cancer (mesothelioma) has gained significant interest. However, reports on adverse effects of NM on ILCs and other components of the serosal immune system are scarce or even lacking. As ILCs are crucial in the first defense against pathogenic viruses and bacteria, it is possible that serosal exposure to NM may lead to a reduced resistance against pathogens. Additionally, affected serosal lymphoid tissues and cells may disturb adipose tissue homeostasis. This review aims to provide insight into key effects of NM on the serosal immune system.

## 1. Introduction/Background

The body has two large coelomic cavities, the thoracic and abdominopelvic cavities, which in turn are subdivided (Figure 1). They may appear well shielded from environmental exposure (inhalation or ingestion) to nanomaterials (NM), but inhaled particles including NM can be translocated to the thoracic cavity when not cleared already by the respiratory mucosal defense [1,2,3,4,5]. It is unclear if NM can reach the peritoneal space upon oral exposure. Ingested NM have been found in abdominal organs like the liver and spleen, most probably via the blood or via the lymphatic system after intestinal uptake [6], based on titanium dioxide NM. NM can be brought intentionally into these cavities, for medical purposes. Medical applications of NM include drug delivery, diagnosis, and anticancer therapy. The inhalation route is used as a non-invasive means for systemic delivery of NM and treatment of local (lung) diseases [7]. However, there are significant gaps in safety aspects, for example regarding the influence of pre-existent lung disease on NM efficacy. This is certainly not a hypothetical concern: exposure of 3D bronchial epithelial models from healthy and asthmatic individuals to metal NM revealed that tissue from asthmatics scored less in tissue defense and detoxification genes than healthy tissue [8]. Additionally, largely unknown is the interaction with local lymphoid elements. Intraperitoneal drug delivery by NM can be superior to systemic drug delivery, because of higher local concentration and less of the normally expected side effects [9], but again it is important to understand if and how they react with the local lymphoid tissues and cells.

The cavities house—in addition to lymph nodes—special lymphoid clusters and many specialized single immune cells, which act at the interface of innate and adaptive immunity [10,11,12,13]. The pleural, pericardial, and peritoneal cavities are lined by serosal membranes with immune-active mesothelium and contain adipose tissues, which have a key role in innate [14] and adaptive immunity [15]. Together they form the serosal immune system. NMs are continuously being improved for drug delivery, screening of various diseases, and tissue engineering. However, effects of (engineered) NM on serosal immune cells and serosa-associated lymphoid clusters have not or hardly been investigated. 

Serosal lymphoid clusters: The lymphoid clusters in the thoracic and peritoneal serosal membranes are called milky spots (MSs; mainly located in the omentum) or fat-associated lymphoid clusters (FALCs), because they are embedded in the adipose tissues. MS and FALC may differ based on location and ontogeny [12], but they have probably similar roles. Moreover, these clusters are found in the translucent (adipose-free) parts of the serosa as well. Therefore they can be grouped together as serosa-associated lymphoid clusters (SALCs) [16] (Table 1). SALC are mainly found in omentum, pericardium, and mediastinum, and considered to be secondary lymphoid organs although their ontogeny is driven by other molecular events and cells than lymph nodes (LNs) [11,12,13]. SALC in the omentum can function in the absence of spleen, LN and Peyer’s patches (PPs): They generated T-dependent B cell responses and memory cells to peritoneal antigens in immune-deficient SLP-mice (lacking Spleen, Lymph nodes and Peyer’s patches) [11].

A remarkable feature of SALC is the presence of many innate lymphoid cells in addition to conventional lymphocyte subpopulations [12,17,18,19,20,21,22]. SALC are described by [23] as “hubs that are important for providing a second line of defense between the mucosal surfaces and a systemic immune response, working to compartmentalize antibody mediated immune responses”. Are peritoneal and thoracic cavities indeed compartmentalized and if so, to what extent? The different cavities start as one coelom in embryonic life, but after birth there is no bulk flow between the pleural and peritoneal cavities [24], and co-operation between the cavities is perhaps restricted to “overload” conditions in a specific cavity [25].

Immune cells in the thoracic and peritoneal spaces: Understanding lymph draining patterns in the thoracic and peritoneal cavities is important to understand the fate of immune cells and engineered nanomaterial (ENM) in these cavities. The tiny space between the cavity wall and the organs is filled with a lubricating fluid. The left side of the thorax and the lower part of the body including the abdominal organs, drains onto the thoracic duct, which carries lymph and emulsified fats known as chyle. Intraperitoneal injected fluorescent tracers are able to reach the mediastinal lymph nodes via the thoracic duct [27]. The right side of the thorax (right side of heart, right lung, major part of mediastinum, and diaphragm) drains to the right lymphatic duct. The slippery fluids contain detached mesothelial cells and free-floating immune cells. A large part of the free B lymphocytes are non-conventional innate B1 cells (sIgM^hi^, sIgD^low^, and CD11b^+^) (Table 2). They produce polyclonal natural antibodies, often carbohydrate-specific, which can respond quickly in the early stage of an infection. Interestingly, there is a sex-related difference in cell content in the pleural space of the rat [28]; Figure 1B in [16]. The free immune cells patrol the cavities, enter and leave SALC, and depart via “holes” in the serosa called stomata to move to LN [29,30]. These so-called stomata appear to be unique to the parietal pleura [31]. They are present on the diaphragm of several species, albeit only or mainly at the peritoneal side [1,32]. In rodents, the presence of stomata at the pleural side might be restricted to the retrocardiac pleural folds, which easily collapse onto the diaphragm [16].

Mesothelium: Mesothelium forms the outer lining of the serosa; it plays a role in serosal tissue homeostasis and repair, and in immunity [35,36,37]. Mesothelial cells synthesize and secrete a host of mediators upon external signals, initiate and regulate inflammation, recruit inflammatory cells, actively transport particulates across the serosal membrane, and present antigen to T cells. The latter roles resemble those of the microfold (M) cells in the mucosa.

Mesothelial cells can undergo transition from more epithelium-like to more mesenchymal-like cells described as epithelium to mesenchymal transition (EMT) or mesothelium to mesenchymal transition (MMT) [38]. In EMT, epithelial cells have been shown to lose cell–cell junctions by downregulating E-cadherin and other junctional proteins and reduce attachment to the basal lamina. Expression of various mesenchymal cell-associated markers confirms the change into mesenchymal cells. Mesothelial cells can undergo (some of) these changes as well, and, after reducing attachment to the basement membrane become free-floating, macrophage-/mesenchymal-like cells in the peritoneal and thoracic fluid. Free-floating mesothelial cells can migrate into other organs like the lungs, where the transcription factor Wilms tumor-1 (WT-1)+ mesothelial cells transform into bronchial smooth muscle cells, vascular smooth muscle cells, and fibroblasts. In this way, mesothelial cells possibly contribute to lung fibrosis. Jackson-Jones et al. described the changes in molecular expression of mesothelial cells, which cover omental SALC, upon exposure to zymosan [39].

Adipose tissue depots: Most of the abdominal adipocytes are “white” (WAT; cells with one fat droplet; and nutrient storage). Small depots of “brown” adipocytes (BAT; cells with multiple small fat droplets but overall less stored fat than white adipocytes; and a role in thermogenesis) can be found in the thorax. “White” adipocytes can acquire a more “brown” characteristic and is then called “beige” (cells with several fat droplets). BAT and “beige” fat are less prone to inflammation in comparison to WAT and express less inflammatory genes [40,41]. Most information on the role of adipocytes in immunity is obtained from WAT, but can generally also apply to BAT and “beige” fat [42,43]. Adipocytes produce and secrete cytokines or adipokines such as IL-6, the tumor necrosis factor, adiponectin, and leptin. Leptin modulates adipocytes via, e.g., Toll-like receptor expression [14]. They also express multiple Toll-like receptors to recognize pathogens and to initiate immune responses, and they can secrete a variety of monocyte/macrophage chemoattractant molecules [14,44]. Adipocytes and macrophages interact closely, for example via adipocyte-produced IL4 and macrophage-produced norepinephrine, which leads to beige fat formation. Interestingly, the role of the macrophages in adipose tissue homeostasis is related to type 2 immunoregulatory phenotype (CD206^+^CD301^+^IL-10^+^) and strictly regulated by type 2 innate-like lymphoid cells (ILC2s) and eosinophils [45,46]. Adipose tissues have been proposed as a source of stroma for visceral (abdominal and thoracic) lymph nodes and SALC. Lymphotoxin-b receptor signaling and NF-kB2-RelB signaling pathway activation blocked maturation of adipocyte precursor cells and instead promoted lymphoid tissue stromal cell differentiation [47].

Scope of the review: The thoracic and peritoneal cavities are protected by a serosal immune system, which repairs tissue damage, preserves serosal and adipose tissue homeostasis, and at the same time cooperates with the mucosal immune system to maintain mucosal integrity. These important roles of the serosal immune system may be affected by exposure to (E)NM, but there is almost no information about their effects on SALC and serosal single immune cells. While there is an extraordinary diversity of (E)NM, this review aims to describe common features of NM exposure on the serosal immune system.

## 2. NM and the Thoracic Cavity

### 2.1. Translocation Route of Inhaled NM to the Thoracic Cavity

A considerable part of inhaled NM deposit in the alveoli. They can remain there or can be transported to the draining lymph nodes, to organs like the liver, kidneys, spleen, and heart, to the chest wall and diaphragm/parietal serosa [3,48] and into pleural fluid [49]. How NM are translocated to the chest wall, diaphragm, and into pleural fluid is still not clear, but might occur by NM-loaded macrophages traveling via pleural lymphatic drainage or might go via blood capillaries [50]. NM may also enter the thorax space directly, whether or not as part of a local inflammatory process: NM were observed penetrating the lung pleura (multiwalled carbon nanotubes in rats and mice) [48,51,52]. It is important to realize that the pleura of rats are impressively thinner than the pleura of humans. Agglomerated multiwalled carbon nanotubes appeared to penetrate alveolar septa as well (Figure 3 in [53]), supporting the observation that even agglomerated NM can penetrate tissues. A review of kinetics of NM, mainly nanotubes, is given by [54].

### 2.2. General Effects in the Thoracic Cavity

The serosa and the thoracic fluid velocity can be affected in response to particle exposure [3,16]. Serosal inflammation may occur: particles can induce granulomata and fibrotic plaques [1]; reversible pleural mononuclear cell aggregates were observed after inhalation of multiwalled carbon nanotubes [55]. Flat mesothelial cells can become cuboidal, release inflammatory mediators, can start to express macrophage markers, and detach from the basement membrane and become free-floating in the pleural and pericardial fluid. The number of leukocytes and levels of cytokines increase in the pleural cavity lavage [56]. Fluid velocity accelerates by widening of already opened stomata and opening of formerly closed stomata. In addition, there is increased drainage of the pleural interstitium and there is also increased diffusion and transcytosis. NM and associated inflammation may block stomata and disturb drainage. In female workers, inhalation exposure to polyacrylate nanoparticles was associated with pleural watery content, pleural granulomatous inflammation and NM were observed in the mesothelium [57]. The mesothelium can proliferate and this may lead to mesothelioma when it is extensive and atypical. Up until now, the association between NM and mesothelioma is unclear [3]. It is postulated that activation of the NLRP3 inflammasome may play a role in the pathogenesis of mesothelioma [58]. Other involved molecular events were reviewed by [59].

Therapeutic intrapleural instillation of talc microparticles has been used in the treatment of pleural effusions, but can also result in respiratory failure. This may depend on the size of talc particles and therefore, the inflammatory response to intrapleural instillation of small particles was compared to the instillation with larger microparticles in rabbits. The instillation of especially smaller particles caused a systemic (C-reactive protein and IL8) and lung response suggesting that in order to reduce side-effects, larger talc particles should be preferred to reduce the chance of migration to the pulmonary parenchyma [60].

## 3. NM and the Peritoneal Cavity

Intraperitoneal drug delivery represents an attractive strategy for the local treatment of cancer cells of organs confined to the peritoneal cavity (e.g., ovary, colon, and pancreas) to maximize local therapeutic efficacy while limiting systemic side effects [61]. As such, NMs are intentionally administered to the peritoneal cavity as part of chemotherapeutic treatment or diagnostic imaging [62,63]. A major disadvantage is the rapid clearance of NM from the cavity via (1) stomata and lymphatics into the thoracic duct and (2) direct absorption through the peritoneum. This clearance could be prevented by coating NM, but this lead to adherence of NM to the serosa [64]. Unfortunately, the serosa were not examined for potential adverse effects. Microparticles and aerosolized NM are less easily cleared by the lymphatic system, but they are not uniformly distributed throughout the cavity due to gravity. Accumulation of the microparticles and aerosolized NM at the lower part of the cavity caused serosal (granulomatous) inflammation and adhesions [62,63]. Iron from intraperioneally-injected iron-containing NM accumulated in the liver, spleen, and mesentery but the highest accumulation was observed in the omentum [65], which is the most common metastatic site for ovarian cancer [66]. Another strategy to increase the residence time of NM in the peritoneal cavity is to use biodegradable hydrogels containing NM for a sustained release [67,68,69].

## 4. Effects of NM on Serosal Lymphoid Clusters and Immune Cells

As stated in the Introduction, effects of NM on serosal immune cells and SALCs have not or hardly been investigated; there is some information on particles of unknown size or microsize, but most information comes from infection models. Results obtained from microparticles may not apply directly to NM, also because NM acquire a “protein corona” in the thoracic and peritoneal fluids due to the adherence of host proteins to the NMs surface. It is well known that formation of a protein corona can alter NM uptake by macrophages and possibly other cells, such as mesothelial cells. The protein corona can be manipulated to enhance drug delivery and anticancer activity [70], but also to improve immune safety. An extensive review of molecular and immunotoxic effects of NM on immune cells in general is given by [71].

### 4.1. SALC

Particles can accumulate in SALC, leading to, for example, the black spots on the parietal pleura of individuals living in urban areas (environmental ultrafine and fine particles; anthracotic SALC) [72], and hyaline pleural plaques, associated with exposure to asbestos (anthracotic SALC or inflammatory processes on the parietal pleura as a result of blockage of the stomata by asbestos fibers) [73]. Induction and enlargement of existing SALC have been linked to inhalation or instillation of particles in experimental animals, but an extensive description of changes, like formation of germinal centre-like structures, and molecular events in SALC has been restricted largely to experimental microbial and nematode infection models. Peritoneally transfected labeled-SRBCs accumulated within 2 hrs in omental SALC [11]. Most of these SRBCs were taken up by CD11b^+^ cells. The SRBC also increased the cellularity in the omentum, especially the CD11c−CD11b^hi^ population. Activation of immune cells in the peritoneal space by infectious agents stimulated their migration into the omentum. The responses in the cavities appear to stay confined to the site of infection: Omental SALC reacted much more than pleural SALC upon percutaneous injection in mice with *Schistosoma mansonii*, a nematode with a preference to the mesenteric veins [74], whereas a subcutaneous injection with *Litomosoides sigmodontis*, a nematode that resides in the pleural cavity in its early stage, activated FALC/SALC in the thorax, but not in the peritoneum [75]. In addition, IgM increased in pleural lavage fluid, but not in the peritoneal space nor in serum, and *Litomosoides*-specific IgM was secreted especially by B cells in thoracic SALC.

### 4.2. Single Immune Cells

Effects on single cell numbers of (micro)particles are reported mostly in older literature. Lehnert and Tech instilled polystyrene microparticles in the lung [76]. This caused a five-fold increase in polymorphonucleated leukocytes (PMNs), and a two-fold increase in alveolar macrophages (AMs). No increase in pleural fluid PMN was found, but pleural fluid macrophages increased, followed later by a clear increase in the pleural mast cell and fewer lymphocytes. Pleural lavage fluid was studied in mice, at 30 min and up to 7 days after the intrapleural instillation of silica or tungsten microparticles. Leukocyte counting in the lavage fluid showed that initially total number of pleural cells decreased, followed by an intense increase in granulocytes and monocytes/macrophages. The morphology of the macrophages was changed distinctly. Both types of particles induced the complete disappearance of mast cells from the pleural space [77].

## 5. Serosal–Mucosal Interaction

The lung mucosa has an extensive immune barrier to deal with inhaled airborne material and a disturbance of this barrier has serious consequences. It is long thought that inhalation of (agglomerated/aggregated) NM leads to clearance from the lungs via more or less simple innate mechanisms, namely uptake by macrophages, mucociliary clearance, and/or drainage to local lymph nodes. However, the interaction of NM with mucosal defense appears much more complicated with innate and adaptive immune cell activity, and may resemble how pathogens and antigens act in the lung (elegantly explained by [78]). In short: Inhalation of NM leads to an early response against particles, regardless particle characteristics. First, type 1 (injury—ILC1, M1 Th1, and phagocytosis) and type 2 (tissue repair—ILC2, M2, and Th2) acute inflammation is induced. In the course of a few days and depending on particle characteristics, concentration, and exposure duration, this inflammation becomes regulated by the balance between activation of Type 3 responses (ILC3 and Th17), Tregs, Bregs, and myeloid-derived suppressor cells. Imbalances and dysregulation of the process can lead to lung and pleural granulomatous inflammation, progressive fibrosis, autoimmune dysfunction, and cancer.

As described in the Introduction, SALC (together with immune cells in the fluid of the spaces and in the visceral adipose tissue in the cavities) provide a second line of defense for the mucosae [10,15]. When the intestinal mucosal barrier is affected, B1 B cells in the peritoneal cavity use molecular signals like CXCL13, integrins, and CD9 to migrate from the peritoneal space to the omentum, mesentery, and spleen. The omentum contains the conventional B2 B cells as well and is able to contribute to systemic humoral immunity. It is largely unknown if B1 B cells in the mediastinum follow similar pathways to protect the respiratory tract mucosa. SALC and omentum may possibly also serve as a source of (precursor) cells for the mucosae. B1 B cells in murine omentum could be precursors of part of the plasma cells in the lamina propria of gut mucosa [79], whereas ILCs in SALC and visceral adipose tissue depots might act as a source for ILCs in the mucosae. In which way the innate lymphocytes in mucosae and the thoracic and peritoneal cavities interact with each other needs still to be explored, despite a tremendous increase in information on these cells [80]. The influence of the serosal immune system on the mucosal immune system is especially prominent in conditions like obesity and metabolic syndrome. Immune dysregulation in obese subjects can result in a chronic low-grade inflammation in which visceral adipose tissue, especially the omental depot, is thought to play a key role. It is characterized by increased infiltration and activation of innate and adaptive immune cells, which in turn leads to a shift from predominantly Type 2 cells to T1 cells and a decrease in Tregs in the adipose tissue [81,82]. Nanomaterials play an important role in the treatment of obesity and obesity-related metabolic diseases [83]. NM could treat obesity and obesity-related metabolic diseases by improving intestinal health and reducing energy intake (used as food fillers, as enhancers of probiotic bacteria in intestinal microflora, and/or lipase inhibitors in the stomach). Moreover, as obesity is usually accompanied by various abnormalities of cells (adipocytes, macrophages, and vascular endothelial cells), NM can treat such cell abnormalities by regulating redox homeostasis (which would reduce the mitochondrial oxidative stress disorders) and removing free lipoprotein in the blood (by apolipoprotein mimetic peptide particles). Moreover, NM can be used for the detection of obesity-related factors (such as adipokines and inflammatory cytokines) and the diagnosis of metabolic diseases such as atherosclerotic plaques [83].

NM might not need to enter the cavities to induce cellular immune changes there [11]. Intranasally administered influenza virus in C57BL/6 mice induced recirculating antigen-specific CD8+ T cells in the peritoneal space. The authors found comparable results with an orally administered intestinal helminth infection in a transgenic mice strain. Both studies together showed that antigen-experienced CD4+ and CD8+ T cells recirculate through the peritoneal cavity (and omentum), even when infection originally occurred outside the peritoneal cavity. Of course, infectious agents may behave differently from NM, as NM do not possess antigenic structures and do not multiply, but the differences are possibly less expressive than previously thought [78]. Jackson-Jones et al. [75] showed that fungus-induced airway inflammation in mice lead to activation of B cells in pleural SALC. Thus, stimulation of immune cells elsewhere, especially in the mucosa, may have consequences for the serosal immunity.

Of course, infectious agents may behave differently from NM, as NM do not possess antigenic structures and do not multiply, but the differences are possibly less expressive than previously thought [78]. Jackson-Jones et al. [75] showed that fungus-induced airway inflammation in mice lead to activation of B cells in pleural FALC/SALC. Thus, stimulation of immune cells elsewhere, especially in the mucosa, may have consequences for the serosal immunity.

## 6. Discussion/Conclusions

Engineered NMs are increasingly used for therapeutics and diagnostics. Part of these ENMs are administered in the peritoneal and thoracic cavities. At the same time there is a growing interest in these cavities, because of translocation of inhaled (nano)particles from the lungs to the thoracic cavities [1,84] and obesity-related health issues such as low-grade inflammation and associated immune dysfunction in the cavities’ (visceral) adipose tissue depots [41,85] (Figure 2). The cavities house specialized lymphoid tissues and immune cells, which in cooperation with the immune roles of its adipose tissue depots and serosal lining forms the serosal immune system.

The serosal immune system is well known for taking care of local tissue repair and homeostasis of the serosa and the adipose tissue depots, but our understanding of its role in humoral and cellular immunity is still limited. Studies with infection models have shown that it is heavily involved or perhaps even central to innate immunity, for example in the production of polyclonal IgM via ILC2 stimulation [75].

The system cooperates with the mucosal immune system, as infection at mucosal sites induces responses in the cavities and vice versa changes in serosal immune cells have effects on mucosal immunity [11,75].

Despite its apparent importance, the effects which engineered NM may have on the serosal immune constituents have not been investigated sufficiently yet. Studies with inhaled nano- and microparticles and with infectious agents have indicated that meaningful effects can be expected indeed. Adverse cellular responses have been observed upon exposure to NMs [86], and currently several hypotheses regarding how NMs induce adverse cellular effects exist: (i) via oxidative stress leading to proinflammatory effects [87], (ii) through genotoxicity [88], (iii) via NM dissolution, i.e., release of potentially toxic ions and/or other constituents [89,90], and (iv) via the fiber paradigm involving nanofibers in particular those with the specific characteristics of high rigidity and high aspect ratio NMs of which it was shown that multiwalled carbon nanotubes caused granulomas in the peritoneal cavity [3,91]. The majority of NM studies have been performed under healthy conditions. Therefore, it is not clear whether or not the effects are changed under diseased conditions, for example if the barrier function of the peritoneum is altered, if the activity of macrophages in the cavities are changed or if changes occurred in the content and composition of proteins in the cavity fluids, which might bind and alter the biological activity of NM as mentioned previously.

Although NM are increasingly used in the field of medicine, it is important to realize that inhalation of seemingly inert NM may provoke immune responses in the respiratory mucosa, which resemble those induced by pathogens and antigens, and which—depending on host characteristics and NM characteristics, dose and/or exposure duration—may lead to imbalance and dysfunction of mucosal and serosal immunity [78]. Comparable events may occur at the intestinal mucosa and in the peritoneal cavity. Therefore, safety evaluation of engineered NM should take the potential adverse effects on both the serosal and mucosal immune functioning into account, starting with investigation of the performance of innate immunity including ILCs.

## Figures and Tables

**Figure 1 ijms-22-02610-f001:**
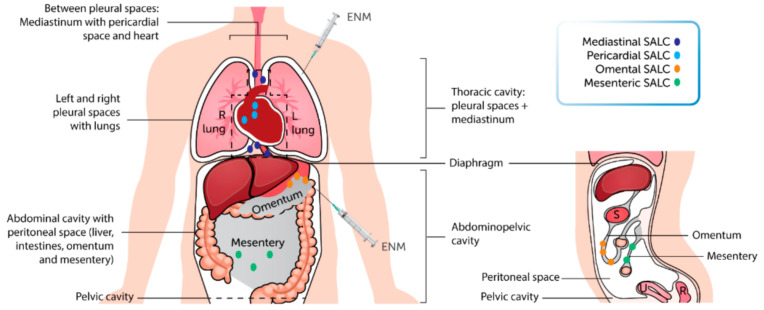
Subdivision of the two coelomic cavities: the thoracic and abdominopelvic cavities. R: rectum; S: stomach; and U: uterus.

**Figure 2 ijms-22-02610-f002:**
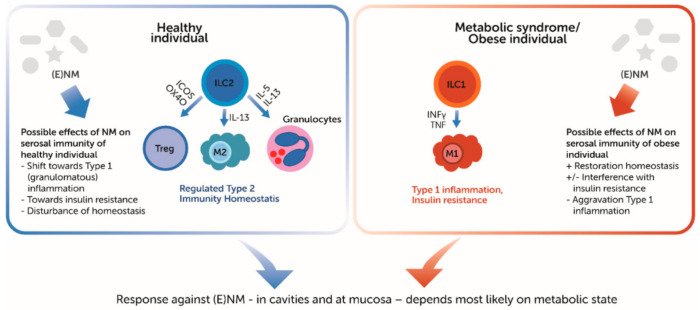
Regulation of visceral adipose immune function in health and metabolic disease. ICOS: inducible costimulatory molecule; ILC: innate lymphocyte; Treg: regulatory T cell; M: macrophage. Based on Figure 1 in [23].

**Table 1 ijms-22-02610-t001:** Characteristics used to distinguish milky spots from fat-associated lymphoid clusters, grouped together as serosa-associated lymphoid clusters.

	Serosa-Associated Lymphoid Clusters (SALCs)		
	Milky Spot (MS) Synonym: omFALC [12,17,18,19,20,21,22]	Fat-Associated Lymphoid Cluster (FALC)	Lymphoid Cluster, not Embedded in Adipose/Fat Tissue
**Ontogeny**	Develop independent of ILC3/LTi cells and CCL19 and CCL21, but are defective/absent in Cxcl13^−/−^ and Ltα^−/−^ mice Observed at 35 weeks of gestation in humans [12,17,18,19,20,21,22]	FALCs develop independent of ILC3/LTi cells and LTβR signaling, but depend on TNF signalling on stromal cells, IL-4R signalling and invariant natural killer T cells [12,17,18,19,20,21,22] Mesenteric FALC are formed after birth	Possibly MS in translucent areas of the serosal membranes
**Location**	Especially in greater omentum Prominent at adipose tissue locations, but also in translucent areas of the serosal membranes [26]	In omentum, mesentery, pleural and pericardial serosal membranes	In omentum [26], and rodent retrocardiac pleural fold [16]
**Function**	Central role in innate B cell maintenance and activation [19]		Possibly identical function, if they are similar to MS in translucent areas
**Microanatomy**	Always covered by mesothelium Often, but not always embedded in adipose tissue	Always covered by mesothelium Embedded in adipose tissue	Always covered by mesothelium Not embedded in adipose tissue

**Table 2 ijms-22-02610-t002:** Overview of innate lymphocytes in visceral adipose tissue, SALCs/FALCs, and cavities ^1^.

Types of Innate Lymphocytes (ILCs)	Types of Innate B Cells (IBCs)
*Non-cytotoxic Tbet-dependent ILC1 (NK)*	*B1a cells CD5^+^, CD11b/Mac1^+^* Responses are T cell-independent Cells have high production of natural antibodies B1a antigens are often carbohydrate- and rarely protein-specific Antibody isotype is IgM and antibody avidity is low Cells develop in foetal liver and are self-renewing in situ
Secrete TNFalpha and INFgamma Are key to control early (viral) replication at initial sites of infection	
*GATA3-dependent ILC2*	*B1b cells CD19^hi^CD5^lo^CD11b^hi^* Responses are largely T cell-independent Natural antibody production poorly investigated B1b antigens are possibly carbohydrate- and protein-specific Antibody isotype is IgM and antibody avidity is low Cells develop in foetal liver and are self-renewing in situ Cells share many characteristics with marginal zone B Cells but differ in distribution and B cell receptor (BCR) signalling pathways
Secrete IL-5 and IL-13 and are regulators of Type 2 immune cells to maintain WAT homeostasis Are involved in browning of WAT, leading to beige adipose tissue	
*RORgamma T-dependent ILC3*, including LTi-cells	
Secrete IL-17A and IL-22 Can convert into ILC1 cells Required to initiate antibacterial actions of epithelial cells	
*Id3-dependent ILCregs*	
Secrete IL-10 Suppress ILC1 and ILC3 activity Regulatory role in intestinal homeostasis and innate immune defences, like Treg cells	

^1^ Based on [16,23,33,34].

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
