# Peer review of "Nanomaterials and the Serosal Immune System in the Thoracic and Peritoneal Cavities"

_ijms, 2021, doi:10.3390/ijms22052610_

Round 1

Reviewer 1 Report

The authors want  to highlight a potential  and to date poorly investigated risk of nanomaterials (NMs)  in impacting innate immunity in serous cavities and visceral fat. They  draw conclusions from the pathogenic mechanisms of infectious agents and inhaled nanoparticles. 

The authors do not introduce to the reader the significance and importance of nanomaterials in modern therapy. The authors do not explain or introduce their therapeutic strength and important articles that have already described in animals the effect of nanomaterials on myeloid activation and macrophage polarization and neutrophil NETs formation that leads to capillary obstruction ect. The authors discuss that proteins adhere to nanomaterials, but they do not discuss that this quality is used for example to remove metastases (ovarian cancer) from the thoracic cavity.

The authors mention that NMs may influence the visceral  adipose immune response. They do not explain why.  NMs may potentiate adipose inflaming through M1 macrophage polarization for example. What is the importance of NM therapy in obesity? NMs are used to treat obesity, they remove lipoproteins in blood, improve intestinal health and reduce energy uptake ect.

The authors should evaluate how the negative side effects of NMs may reduce the therapeutic opportunities of NMs. They should discuss experimental approaches needed to address this topic.

The review in its present form is hard to understand for a reader that has not been introduced to NMs and their effect on the immune response and their importance in novel therapeutic strategies. The review does not discuss  novel key literature that has shown important effects of NMs on myeloid activation  and does not relate these findings to serous cavity and visceral adipose homeostasis .

Author Response

Reviewer 1

1A. The authors do not introduce to the reader the significance and importance of nanomaterials in modern therapy. The authors do not explain or introduce their therapeutic strength and important articles that have already described in animals the effect of nanomaterials on myeloid activation and macrophage polarization and neutrophil NETs formation that leads to capillary obstruction ect.

Response. The Introduction addresses in short the use of nanomedicine, its advantages and disadvantages, as far as the topic of the present manuscript is concerned. In order to concentrate on the topic of NM and serosal immunity, the effects of NM in general on the immune system, including on macrophages, has not been described. Instead, the reader is referred to a review (lines 241-242, under section 4: ‘An extensive review of molecular and immunotoxic effects of NM on immune cells in general is given by [71]’).   

1B. The authors discuss that proteins adhere to nanomaterials, but they do not discuss that this quality is used for example to remove metastases (ovarian cancer) from the thoracic cavity.

Response. A description and reference to manipulation of protein corona for drug efficacy has been given in section 4, lines 239-240.

2B. What is the importance of NM therapy in obesity? NMs are used to treat obesity, they remove lipoproteins in blood, improve intestinal health and reduce energy uptake ect.

Response. An extensive description is included in the section ‘5. Mucosal-serosal interaction’, lines 311-321. 

  1. The authors should evaluate how the negative side effects of NMs may reduce the therapeutic opportunities of NMs. They should discuss experimental approaches needed to address this topic.

Response. The discussion now includes a description of negative side effects (starting at line 365) and the conclusion at the end of the discussion recommends to start with the investigation of innate immunity, including ILCs.

  1. The review in its present form is hard to understand for a reader that has not been introduced to NMs and their effect on the immune response and their importance in novel therapeutic strategies. The review does not discuss novel key literature that has shown important effects of NMs on myeloid activation and does not relate these findings to serous cavity and visceral adipose homeostasis.

Response. The possible effects of NM on serosal immune function and metabolism/obesity have been outlined in the new figure (Figure 2), and is addressed in the discussion; the Introduction now addresses the use of nanomedicines (starting at line 36).

Reviewer 2 Report

The authors have described serosal immune system in the two large coelomic cavities. Innate lymphoid cells have interactions with various immune cells and cytokines and play important roles in adipose tissues. This review is expanded upon your review in 2018 (Toxicol Sci.) and newly focused nanomaterials (NM). Few papers described about adverse effects of NM on ILCs. This review would be of assistance for the area of the serosal immune systems. Overall, the manuscript is well written. However, tables are less information about the two coelomic cavities and innate lymphocytes. Graphical representation should be used referring to various reviews.

Comment:

  1. The authors picked several key words up. However, explanation was hardly any FALC (fat-associated lymphoid clusters) and MS (milky spots) in this. Do you refer #13 report (Toxicol Sci, 2018)? When it is illustrated, FALC in adipose function or FALC formation with immune cells including stroma cells would be better representation.
  2. Related comment 1), it is recommended to show by graphical scheme or partially modified figures as follows: ILC2 (reference No. 20; Figure1), FALC (reference No. 33), Section 5 Serosal-Mucosal interaction (Type1 to 3 immunity/inflammation in the lung; reference No. 71, Figure1-3)
  3. In all cell surface markers and Ig types, expression levels represented with plus (+), high (hi), or low (lo) should be placed (changed) superscript characters. Ex: line 75 (sIgMhi, sIgDlow, CD11b+)---change to (sIgMhi, sIgDlow, CD11b+), line119 (CD206+CD301+IL-10+), Table 2 (B1 B cells; CD19hiCD5loCD11bhi)
  4. In page 8, where is “Graphical Abstract”? This might be an answer to my comments 2 and 3.

Author Response

Reviewer 2

The authors have described serosal immune system in the two large coelomic cavities. Innate lymphoid cells have interactions with various immune cells and cytokines and play important roles in adipose tissues. This review is expanded upon your review in 2018 (Toxicol Sci.) and newly focused nanomaterials (NM). Few papers described about adverse effects of NM on ILCs. This review would be of assistance for the area of the serosal immune systems. Overall, the manuscript is well written.

  1. However, tables are less information about the two coelomic cavities and innate lymphocytes. Graphical representation should be used referring to various reviews.

Response. A Figure (Figure 1) has been prepared to replace Table 1.

Comments:

  1. 1. The authors picked several key words up. However, explanation was hardly any FALC (fat-associated lymphoid clusters) and MS (milky spots) in this. Do you refer #13 report (Toxicol Sci, 2018)? When it is illustrated, FALC in adipose function or FALC formation with immune cells including stroma cells would be better representation.

Response 1: Table 1 is now included which summarizes the key issues of FALC, MS and SALC. The term SALC groups together both MS and FALC and lymphoid clusters in translucent areas of the serosal membranes (the last may be the milky spots in translucent areas, as described by e.g. Huyghe et al. 2016).

  1. Related comment 1), it is recommended to show by graphical scheme or partially modified figures as follows: ILC2 (reference No. 20; Figure1), FALC (reference No. 33), Section 5 Serosal-Mucosal interaction (Type1 to 3 immunity/inflammation in the lung; reference No. 71, Figure1-3)

Response 2:

- Ref 23 (= ref 20 in previous version), Figure 1 (beige, brown and white fat): The authors thank the reviewer for this suggestion. A figure has been prepared, based on the suggested figure in ref 23 (also based on personal communication with Huyghe, from ref Huyghe et al. 2016).   

- Ref 39 (= ref 33 in previous version): Figure on FALC/SALC/MS. The literature on MS and FALC is not unambiguous with respect to differences between MS and FALC. Therefore, the authors thought that a Table could better deal with these differences than a Figure. 

- Ref 78 (= ref 71 in previous version). A significant part of the graphic summary was based on figs 1-3 from ref 78 (Ma 2020), which unfortunately, was not included in the manuscript, but submitted in a separate file. To avoid any inconveniences for the reviewers, we now included the graphic summary in the manuscript.

  1. In all cell surface markers and Ig types, expression levels represented with plus (+), high (hi), or low (lo) should be placed (changed) superscript characters. Ex: line 75 (sIgMhi, sIgDlow, CD11b+)---change to (sIgMhi, sIgDlow, CD11b+), line119 (CD206+CD301+IL-10+), Table 2 (B1 B cells; CD19hiCD5loCD11bhi).

Response 3: The characters ‘hi’, ‘low’, ‘+’, and ‘-‘ have been placed in superscript throughout the manuscript.

  1. In page 8, where is “Graphical Abstract”? This might be an answer to my comments 2 and 3.

Response 4: The graphic summary was submitted separately from the main manuscript and was apparently not available to the reviewer. It has now been included in the main manuscript. Because the graphic summary may not sufficiently address the reviewer’s comments 2 and 3, an additional Figure was included on NM interaction and different metabolic states, in addition to a Table summarizing the potential differences and similarities of FALC, MS and SALC. 

Round 2

Reviewer 1 Report

The authors have addressed in detail all concerns.